# Effect of Size and Shape on Electrochemical Performance of Nano-Silicon-Based Lithium Battery

**DOI:** 10.3390/nano11020307

**Published:** 2021-01-25

**Authors:** Caroline Keller, Antoine Desrues, Saravanan Karuppiah, Eléa Martin, John P. Alper, Florent Boismain, Claire Villevieille, Nathalie Herlin-Boime, Cédric Haon, Pascale Chenevier

**Affiliations:** 1CEA, CNRS, IRIG, SYMMES, STEP, University Grenoble Alpes, 38000 Grenoble, France; caroline.keller@univ-grenoble-alpes.fr (C.K.); krsaro87@gmail.com (S.K.); elea.martin@etu.unistra.fr (E.M.); claire.villevieille@cea.fr (C.V.); 2CEA, LITEN, DEHT, University Grenoble Alpes, 38000 Grenoble, France; johnalper@googlemail.com (J.P.A.); cedric.haon@cea.fr (C.H.); 3CEA, CNRS, IRAMIS, NIMBE, LEDNA, University Paris Saclay, 91191 Gif-sur-Yvette, France; antoine.desrues@cea.fr (A.D.); florent.boismain@insa-lyon.fr (F.B.); nathalie.herlin@cea.fr (N.H.-B.)

**Keywords:** silicon nanoparticles, silicon nanowires, synthesis, high energy density, lithium-ion batteries, high-capacity anode, VLS, laser pyrolysis, size effect, shape effect

## Abstract

Silicon is a promising material for high-energy anode materials for the next generation of lithium-ion batteries. The gain in specific capacity depends highly on the quality of the Si dispersion and on the size and shape of the nano-silicon. The aim of this study is to investigate the impact of the size/shape of Si on the electrochemical performance of conventional Li-ion batteries. The scalable synthesis processes of both nanoparticles and nanowires in the 10–100 nm size range are discussed. In cycling lithium batteries, the initial specific capacity is significantly higher for nanoparticles than for nanowires. We demonstrate a linear correlation of the first Coulombic efficiency with the specific area of the Si materials. In long-term cycling tests, the electrochemical performance of the nanoparticles fades faster due to an increased internal resistance, whereas the smallest nanowires show an impressive cycling stability. Finally, the reversibility of the electrochemical processes is found to be highly dependent on the size/shape of the Si particles and its impact on lithiation depth, formation of crystalline Li_15_Si_4_ in cycling, and Li transport pathways.

## 1. Introduction

High energy density in lithium-ion batteries (LiB) requires active materials with enhanced lithium absorption capacity and a large potential gap. On the anode side, silicon appears as the most promising material to enhance the specific capacity for several reasons [1]. Si is an earth-abundant, low-cost, and non-toxic element, with a mature industrial knowledge base from its application in electronics, optics, and photovoltaics. Furthermore, its low potential, close to Li/Li^+^, makes it suitable for composites with graphite, the current commercial anode material [2,3,4]. Finally, Si shows a high Li-alloying capacity but, consequently, an undesired high volume change in cycling [5]. This deleterious volume change stimulated the development of Si nanomaterials, because Si at the nano scale can withstand swelling without fracture during LiB cycling. This has been demonstrated for a variety of morphologies including Si nanoparticles (SiNP) [6,7,8,9,10,11], Si nanowires (SiNW) [3,12,13,14], Si nanotubes [15,16], and Si porous nanomaterials [17,18,19]. However, nanostructuration can be a costly process that needs to be carefully optimized for a dedicated application.

The size of the nano-Si and its impact on lithiation mechanisms has been the subject of numerous papers. Several in situ TEM investigations have shown that fracture of crystalline SiNWs [20,21] and SiNPs [21,22] occurs above a critical diameter of ≈150 nm. A critical diameter was also predicted by a mechanical numerical model [23] at 90 nm for SiNPs and 70 nm for SiNWs. By contrast, several electrochemical studies in LiB investigated the impact of particle size on Si-based anode cycling for small [6,8] and large [7,8,9] SiNPs, with the best results at diameters of 20–50 nm. As for SiNWs, some studies have suggested the optimal diameter as being around 30 nm [10,24]. These size limits in batteries look much lower than the safe nano-silicon size inferred from calculations or TEM observations. However, these electrochemical studies use different sources of Si with heterogeneous size distributions and within a narrow range of average sizes, making the optimum estimation quite challenging. Additionally, to date, no study has compared different shapes of nano-silicon cycled in the same conditions to enable direct comparison of 0D and 1D structures.

SiNPs and SiNWs can be obtained with a very high control on diameter from well-described synthesis methods, such as colloidal nanocrystal synthesis [25] and chemical vapor deposition (CVD) growth [26,27]. However, these highly controlled growth processes produce insufficient (~1 mg) quantities of material for assembling a lithium-ion battery (requiring >100 mg for typical coin cell electrodes). Typical scalable processes developed for industrial synthesis of SiNPs [28,29] and SiNWs [9,30,31] produce nanomaterial either with a very wide size distribution or with a limited ability to change the diameter. The synthesis of a series of nano-silicon materials of fine-tuned size in a large scale is, thus, a challenge for materials chemists. 

In the present paper, we present optimized, scalable processes to obtain large batches of SiNPs and SiNWs allowing a comparison of Si nanomaterials in shape (spherical or cylindrical) and size. All materials were tested in half-cell Li-ion batteries in the same conditions to investigate the impact of shape/size on the electrochemical performance and reaction mechanisms. 

## 2. Materials and Methods

Chemicals: Calcium carbonate (CaCO_3_ nanopowder, 50 nm), sodium citrate, ascorbic acid, and carboxymethylcellulose sodium (CMC) were purchased from Sigma Aldrich. Gold tetrachloroauric acid (HAuCl_4_), tetraoctylammonium bromide (TOAB), dodecanethiol, and sodium chloride were purchased from Alfa Aesar. Carbon black Super-P (CB) was purchased from Imerys. Diphenylsilane was purchased from Chemical Point. Silane (high purity, 98%) was purchased from Messer.

Silicon nanoparticle synthesis: The silicon nanoparticles (NPs) were synthesized by laser pyrolysis as already described [28,32]. Briefly, a flow of gaseous silane intersects with the beam of a high power CO_2_ laser. In the interaction zone, silane absorbs the laser radiation. Dissociation of the precursor followed by nucleation and growth of Si nanoparticles occurs with appearance of a flame. Key parameters to control the nanoparticle size are the residence time in the laser beam, the nucleation, and the flame temperatures. The main control parameters are the gas flow rates, the dilution of active gases, and the laser power.

Gold nanoparticle synthesis for silicon nanowire growth: 1–2 nm gold nanoparticles (AuNP) are synthetized by the Brust method [33]. Briefly, HAuCl_4_·xH_2_O dissolved in water is transferred in toluene using tetraoctylammonium bromide, then reduced with NaBH_4_ and stabilized with dodecanethiol (dodecanethiol/gold molar ratio 2:1), leading to 1.5–2 nm diameter AuNPs quantitatively. The organic gold nanoparticles are purified by precipitation in ethanol and redispersed in chloroform. Twelve nanometers AuNPs are synthetized using the Turkevich method [34]. Briefly, HAuCl_4_·xH_2_O dissolved in water is mixed with a sodium citrate solution at 100 °C and agitated for 30 min. Higher diameter AuNPs are made following the Ziegler and Eychmüller method [35], by regrowing from the 12 nm AuNPs used as gold seeds. Briefly, a suspension of 12 nm AuNPs in water (120 mL, 90 mg/L) is slowly mixed with an aqueous solution of HAuCl_4_ (60 mL, 3.4 mM), sodium citrate, and ascorbic acid (60 mL, 6.3 and 4.6 mM, respectively). The aqueous AuNPs are purified by centrifugation after precipitation with ethanol. 

Silicon nanowire synthesis: SiNWs are synthetized in a 150 cm^3^ home-built stainless-steel reactor designed to withstand pressure above 50 bars and temperatures above 500 °C. AuNPs are deposited by drop drying either on a NaCl micropowder if the colloid solvent is organic, or on a CaCO_3_ nanopowder if the colloid is in aqueous solution (50 mg AuNPs on 25 g NaCl or 3 g CaCO_3_). After drying, the powder is placed in a 2 cm diameter alumina crucible in the reactor with 12 mL diphenylsilane under vacuum. The reactor is heated to 430 °C within 40 min and kept at 430 °C for 80 min. After cooling, the powder is washed with water if grown on NaCl, or with 4 M aqueous HCl if grown on CaCO_3_, and with dichloromethane. To check for the effect of NaCl or CaCO_3_ on the growth process, some 12 nm AuNPs were transferred in toluene using ligand exchange with hexadecylamine in toluene [36], then replacing hexadecylamine with dodecanethiol and subsequently depositing on NaCl powder by centrifugation. The SiNWs grown from 12 nm AuNPs on NaCl did not show any change in shape and size as compared to the SiNWs grown from 12 nm AuNPs on CaCO_3_.

Materials characterization: Phase identification was performed with powder X-ray diffraction (XRD) technique on a Bruker D8 advance diffractometer θ–2θ configuration with a Cu anticathode (Bruker AXS, Karlsruhe, Germany). The scanning step used was 0.02° with a counting time of 1.2 s per step. Scanning electron microscopy (SEM) was performed on a Zeiss Ultra 55 microscope (Zeiss, Oberkochen, Germany) at an accelerating voltage of 5 kV and working distance of 5 mm. A JEOL 2010 (JEOL, Tokyo, Japan) high-resolution transmission electron microscope (HRTEM) operated at 200 kV was used for TEM and HRTEM observations. For TEM measurements, the powder was dispersed in ethanol and nanoparticles separated with intensive ultrasound using the Hielscher Ultrasound Technology VialTweeter UIS250V (Hielscher, Teltow, Germany). Then, the dispersion was dropped on a grid made of a Lacey Carbon Film (300 mesh Copper—S166-3H). The Brunauer, Emmet, and Teller (BET) method was used to measure the SBET specific surface of the different samples using Micromeritic apparatus Tristar II and Flowsorb 2300 (Micromeritics, Norcross, GA, USA). Electrochemical studies, including electrochemical impedance spectroscopy, were carried out using a Biologic VMP3 multichannel potentiostat (Biologic, Seyssinet-Pariset, France) and an ARBIN charge–discharge cycle life tester (Arbin, College Station, TX, USA). 

Lithium battery assembly and test: SiNPs or SiNWs are mixed in an ink containing 50 wt. % of active material, 25 wt. % of carbon black, and 25 wt. % of CMC dispersed in distilled water. The resultant slurry is coated using the doctor blade method on thin copper foil (12 μm), dried at 80 °C overnight (mass loading 0.19–0.46 mg_si_ cm^−2^, dry thickness around 20 μm), and calendared at 1 ton. A celgard separator was soaked with 150 µL electrolyte of ethylenecarbonate/diethylenecarbonate 1/1 *v*/*v* containing 1M LiPF_6_, 10 wt. % fluoroethylene carbonate, and 2 wt. % vinylenecarbonate. Two thousand and thirty-two coin cells were assembled with lithium metal as reference and counter electrode in an argon-filled glove box and crimp sealed. Electrochemical properties of the half-cells were evaluated in the potential window between 0.01 and 1.0 V vs. Li^+^/Li. All potentials reported below were measured in a half cell configuration in reference to the Li metal counter electrode and are, thus, expressed as vs. Li^+^/Li. The first cycle rate is C/20. Later, cycles are performed at C/5 rate, with a floating time at the end of lithiation until a current of C/100 and C/50, respectively, under 0.01 V. All the capacity values shown in this paper are based on the mass of silicon in the electrode.

## 3. Results

### 3.1. Tuning the Size of SiNPs: Silane Concentration and Residence Time

Batches of SiNPs with different diameters were synthesized by laser pyrolysis. Table 1 presents the main experimental parameters and size measurements of the samples used in this study. The samples are labeled SiNP_X_ where X states the average SiNP diameter. A flow of silane (silicon precursor) is diluted in He with slightly different ratios to control the nucleation and growth. The silane flow rate is varied from 50 to 200 sccm with a total gas flow rate of 1100 ± 100 sccm. Increasing the SiH_4_ to He ratio results in samples with diameters regularly increasing from 30 to 87 nm. The samples SiNP_43_ and SiNP_53_ were synthesized with the same SiH_4_ flow and differ by the He flow rate (1000 vs. 1100 sccm). Finally, in order to reach higher SiNP diameters, we modified the reactor to increase the residence time. The reactant inlet tube close to the laser beam was enlarged from 2 to 4 mm diameter, thus dividing the gas velocity by a factor of 4 and increasing the time of residence in the reaction zone. The synthesis of SiNPs with a diameter of 110 nm could, thus, be achieved, although with a broader size distribution.

The typical morphology and size analysis of the different samples presented in Figure 1 shows the quality of this synthesis approach, with a very narrow size dispersion obtained for most SiNP samples. The SiNP size was measured both from the BET specific area and estimated from TEM images. The BET diameter is consistently larger than the TEM average diameter because of the morphology of the SiNPs forming small agglomerates. The surface area lost in interparticle contacts decreases the measured BET surface and, therefore, increases the estimated diameter. In the same way, the crystallite size deduced from XRD measurements (Appendix A) is always smaller than TEM indicating that the SiNPs are not monocrystalline, as also seen by HRTEM in Appendix A.

### 3.2. Tuning the SiNW Diameter: Catalyst Size and Silane Partial Pressure

Free-standing SiNW growth is achieved by a process similar to the vapor–liquid–solid (VLS) mechanism, using a sacrificial porous support as described in our previous work [30]. The SiNWs are grown at low temperature (430 °C) using gold nanoparticles (AuNPs) as a catalyst. The sacrificial powder of NaCl or CaCO_3_ covered with the AuNPs is heated in a closed stainless-steel reactor with diphenylsilane as a Si source. At the end of the synthesis, the sacrificial template is removed by washing with water or aqueous HCl, respectively. The optimized process yields up to 0.5 g of SiNWs of 10 nm diameter with a low size dispersion, from AuNPs of 2 nm [30].

It has been demonstrated that diameter control in similar CVD processes can be realized by controlling the catalyst size as was demonstrated for carbon nanotubes [37,38] and for SiNWs [39,40]. Mechanistic studies on SiNW CVD growth show that thermodynamic constraints control the nanowire diameter. It was demonstrated that a high partial pressure of silane (90 Pa) at 400–500 °C induces the growth of small diameter SiNWs (2–20 nm), whereas a low partial pressure (5 Pa) at 550–650 °C allows the growth of SiNWs of 50 to 1000 nm diameter [26,27,41,42]. However, our closed reactor does not allow for a direct control on the silane partial pressure.

In our process, the diphenylsilane evaporates then decomposes into silane and tetraphenylsilane, following a disproportionation reaction [43,44,45]. The silane partial pressure is, therefore, controlled by the kinetic balance between its formation from diphenylsilane and its consumption by the SiNW growth. The diphenylsilane disproportionation is slower but starts at a low temperature [44] (from 200 °C), while the silane decomposition on gold is fast but requires a high enough temperature to form the Au-Si eutectic [27] at 363 °C. Thus, a SiH_4_ stock has time to build up in the reactor before the AuNP catalyst turns active for silane decomposition. From diphenylsilane disproportionation rate estimates [44], we can infer a partial pressure of SiH_4_ of 100–500 Pa at the onset of gold catalyzed silane decomposition. At such a high silane pressure, SiNWs can grow as fast as 200 nm/min, and the thinnest SiNWs are favored [26].

We, therefore, developed a novel strategy to achieve size tuning of SiNWs by locally reducing the silane partial pressure. The dense CaCO_3_ nanopowder (50 nm) used as a porous growth support is placed in a cylindrical crucible. The powder compacity generates a gradient of partial pressure from the surface to the bottom of the crucible, as illustrated in Figure 2a. Silane entering the crucible is quickly consumed by the top AuNPs, so that only limited silane can diffuse down in the powder. Therefore, the average silane partial pressure inside the powder is lower than at the surface, which drives the growth of bigger SiNWs if proper catalysts are present. We first demonstrated this effect using a 1 cm^3^ carbon foam cube (Figure 2b). Analysis of the SiNW diameters at the cube surface and deep in the cube show a significant difference of 30%, as presented in Figure 2c. It can be concluded that the chemical rate of silane consumption for SiNW growth is much faster than the rate of silane gas diffusion, although the porosity of the carbon foam is very large (400 µm pores). A much higher silane depletion is expected in the CaCO_3_ nanopowder, in which the pores are thinner (<100 nm) and the pathway more tortuous.

Once able to lower the silane pressure, gold catalysts with the right size are still needed for large SiNW growth. Three colloidal AuNP growth methods [33,34,35] were necessary to access AuNPs with a wide range of sizes. Small AuNPs (1–3 nm) are efficiently grown by dodecanethiol stabilization [33] in an organic solvent, while bigger AuNPs (12 nm) can only be obtained stabilized by citrate [34] in aqueous solution. The latter can be enlarged in a controlled way using an established regrowth method [35] in water in the range 15 to 120 nm (see SEM images of the AuNPs in insets Figure 3a and Appendix A).

We could check the combined effect of AuNP size and thin porosity of the support on the growth of SiNWs from the 12 nm AuNPs (Table 2, Appendix A): SiNWs grown in the CaCO_3_ nanopowder have a diameter of 18 nm, 35% higher than the 13 nm SiNWs grown in the carbon foam. Then, by using AuNP catalysts of increasing size, it was possible to obtain a series of SiNWs of increasing diameter (Figure 3, Appendix A). Note that the growth is not homogeneous, as the thin SiNWs grown on the top constitute a minor population in all samples. This is clearly shown in Figure 3b for SiNW_55_ with a small population at 10 nm, aside the main population at 50 nm. Although numerous, the thin SiNWs represent only a small fraction of the Si volume in the material. Some large SiNWs show a more tortuous shape, and the number of kinks in the SiNWs increases with their diameter. This worm-like morphology, already described in SiNW CVD growth, is typical of an unfavored growth [46,47].

Figure 4a displays the SiNW diameter distribution as a function of the size of the AuNPs used as catalyst. The minor population of small SiNWs is shown with circles. The diameter of the main population of SiNWs nearly matches the size of the AuNPs, as reported for the VLS mechanism [47]. Our strategy for a catalyst-size-directed growth of SiNW, thus, proves efficient, as the SiNW diameter increased over a decade by this method. The distribution width of the main population (measured as the full width at half maximum) is 30–40% for all samples.

The specific area measured by the BET method on the SiNW samples (Figure 4b) drops consistently when increasing the size. The BET and SEM data are in agreement for the smallest SiNWs (SEM estimated specific area of 150 m^2^/g for a BET surface of 194 m^2^/g). However, for larger SiNW samples, they diverge, showing an underestimated count of the small SiNW population by SEM.

### 3.3. Electrochemical Performance in Li-Ion Batteries

The electrochemical performance of SiNPs and SiNWs in lithium batteries are displayed in Figure 5. All materials were tested with 50 wt. % active material in the anode. The quantity of carbon black (25 wt. %) and binder (25 wt. %) were quite high to ensure good electronic conductivity and mechanical stability during electrochemical measurements for all materials. The contribution of the carbon black to the specific capacity, measured independently, is a constant 100 mAh/g. 

The specific capacity is higher and more repeatable for SiNPs (3000–3500 mAh/g) than for SiNWs (2500–3000 mAh/g). All SiNP anodes give a similar specific capacity. For SiNWs, we observe a minimum of specific capacity for the medium size, SiNW_42_. The most favorable trade-off between specific capacity and first irreversible capacity is observed for the largest size, SiNW_55_ (Figure 5c and Appendix A). However, among all samples, only the smallest SiNW anode, SiNW_9_, attains the Coulombic efficiency of 99.5% required for long term cycling (Figure 5d). The lower specific capacity of the SiNWs may be due to difficulties in dispersion of the material during the slurry preparation. The growth in a porous support favors SiNW entanglement, and the obtained 1–10 µm sized agglomerates do not fully separate during slurry process, thus leading to a poorer wetting by the electrolyte in the agglomerates (Appendix A). During the potentiostatic step applied at the end of full lithiation, the equilibrium is not reached, especially for large SiNWs (Appendix A). This indicates residual capacity still available in the SiNW electrode and a kinetic limitation to lithiation in the agglomerates.

The loss in the first Coulombic efficiency (CE) is due to irreversible processes happening in the first lithiation (also called activation). For nano-silicon anodes, it is mostly related to the solid–electrolyte interphase (SEI) formation. It depends on the surface area of the material as demonstrated for graphite particles [48]. Figure 6a presents the first and fifth CE as a function of the BET surface measured for SiNPs and SiNWs. As can be seen for the SiNP electrodes, the excellent correlation shows that the first CE depends linearly on the specific area. The capacity loss in the first cycle is, thus, due to the homogeneous coverage of the silicon surface by a SEI passivation layer. SiNWs follow the same trend, but with a poorer correlation due to a less controlled wetting in the SiNW agglomerates. To the best of our knowledge, this linear correlation has never been reported for silicon. In the subsequent cycles (fifth CE shown on Figure 6a), the CE rises above 95%, and the linear correlation with specific area fades away. On Figure 6b, the first CE is presented as a function of SEM/TEM diameters, *d*, and shows the expected 1/*d* evolution. The SiNW diameter reported here is the average of the most abundant SiNW population. The presence of the small SiNWs brings a large additional surface area and, thus, a higher irreversible capacity in the first cycle. 

Upon long-term cycling, the SiNW_9_ electrodes outperform the other materials due to their enhanced stability and very high CE (Figure 5d). On the contrary, the SiNP specific capacity suffers from a faster fading. The size of SiNPs also has a strong impact on the evolution of the Coulombic efficiency upon cycling. Indeed, for the smallest SiNPs, the CE presents a progressive increase, whereas the larger SiNPs show a fast initial increase then a decrease with a minimum around 10 cycles. This phenomenon could be attributed to an electrochemical sintering [49,50,51] leading to larger SiNPs. Microscopy studies on SiNP anodes in cycling showed general sintering, forming large networks of sintered SiNPs in the anode [49]. Such large Si structures are sensitive to mechanical pulverization in the subsequent cycles, leading to a loss in Coulombic efficiency.

To better assess the impact of the Si shape and size during electrochemical cycling, we plotted the normalized galvanostatic cycle as a function of the cycle number. During the first lithiation (Appendix A), the very long potential plateau at ca. 100 mV is shifted toward lower potential for larger SiNPs, indicating polarization. This is in agreement with a slower Li diffusion along longer distances within large SiNPs [52]. In the following cycles (Figure 7), we can see that polarization during lithiation is increasing as a function of the cycling number for the large SiNPs. For the SiNWs, the potential plateau in the first cycle is also low, indicating anode polarization, which is independent of size (Appendix A), and in the subsequent cycles, polarization does not increase (Figure 8). This discrepancy indicates an influence of the Si shape on the Li diffusion. 

Stronger differences can be seen during the delithiation. For SiNPs, a very long potential plateau at ca. 450 mV appears during the first cycles (second or third depending of the samples) and disappears after more than 30 cycles. This potential plateau is generally ascribed to the biphasic delithiation of the crystalline Li_15_Si_4_ phase [53] and was reported to be size-dependent (appearing with large particles [50,54,55]) or linked to the stress distribution in the case of thin films [56]. The appearance of this plateau indicates that a large part of the Si is fully lithiated to the crystalline Li_15_Si_4_, even if the crystallization of this phase is reported to be more difficult in small SiNPs [50,54,57]. According to the literature, this phase also reduces the Coulombic efficiency [58], which is in agreement with the observed CE evolution of the SiNPs in our study (Figure 5b). 

Thus, the CE fluctuations observed for larger SiNPs would correlate with the pronounced potential plateaus in delithiation, i.e., with the larger amount of crystalline Li_15_Si_4_ formed during lithiation. Deep lithiation has been reported to favor the electrochemical sintering of SiNPs in lithium battery anodes, because the swelling of Si in the form of an amorphous Li alloy induces the formation of necks between neighboring particles [49,51,59]. Within this soft structure, the crystallization of Li_15_Si_4_ in the points of deepest lithiation might strengthen these simple contacts by growing crystals through the necks. Indeed, our SiNPs are connected in a necklace morphology from the growth (Figure 1a), which could enhance this phenomenon.

Surprisingly, this crystalline Li_15_Si_4_ delithiation process is much less visible in the SiNW electrodes, with very small potential plateaus at 450 mV (Figure 8d–f). This shows that the crystalline Li_15_Si_4_ phase is present in a much smaller quantity, or that the lithiated phase remains mostly amorphous [53]. Only the mid-sized SiNW_42_ shows a significant plateau, a fact that can be correlated with its low Coulombic efficiency as compared to the other SiNW samples (Figure 5d). A reason why SiNWs do not undergo electrochemical sintering as easily as SiNPs might be related to their elastic “spring” behavior. Even after grinding and calendaring, SiNWs in the anode remain stiff. Agglomerates of SiNWs contain a nanoscale porosity that was clearly imaged in our recent FIB-SEM study [3]. Such porosity allows swelling during lithiation with a low risk of sintering with neighboring SiNWs. 

## 4. Discussion

The discrepancy in the ability of the nano-silicon material to lithiate into crystalline Li_15_Si_4_ may have a direct impact on the specific capacity retention of the LiB, as shown on Figure 9. The smallest SiNWs show a much higher specific capacity retention over the fourth to 80th cycles, while most of the capacity fading for the SiNPs happens in the cycles 4–30, i.e., when lithiation into crystalline Li_15_Si_4_ is observed.

The influence of the shape/size of the nano-silicon on the reaction mechanisms in cycling and the electrochemical performance can be summarized as follows:First, the irreversible capacity loss in the first cycle increases linearly with the specific area, in a similar trend for SiNPs and SiNWs (Figure 6). This clear correlation indicates that a similar homogeneous SEI layer passivates the silicon surface, independently of Si size and shape.For the SiNPs, the gas-flow process of pyrolysis produces a fine powder, which is easily incorporated in a homogeneous anode and ensures a high specific capacity independent of the SiNP size. The size strongly affects the electrochemical performance, as polarization builds up in cycling over the first 40 cycles. A crystalline Li_15_Si_4_ phase develops in the 2nd–30th cycles, having an impact on the CE behavior and enhancing capacity fade. Polarization, crystalline Li_15_Si_4_ formation, and lower CE might be attributed to electrochemical sintering [49]. The phenomenon is less apparent at lower size, thus small SiNPs should be favored for long term cycling. Alternatively, sintering is efficiently prevented by a carbon coating, conveniently obtained in the same reactor just after pyrolysis, as we recently reported [60].When comparing SiNWs to SiNPs, it appears that the initial specific capacity of SiNWs is lower due to their micron-sized agglomerate form, while the CE of SiNWs is more stable and can be very high for the smallest size. It seems that the 1D shape of SiNWs brings a significant advantage in maintaining a 3D porous structure in the material, because stiff wires do not pack as tightly as particles. This 3D structure has a strong impact both on lithium diffusion and on avoiding Si electrochemical sintering.The effect of size for SiNWs is more complex than for SiNPs. It seems to reflect a trade-off between specific area (decreasing with diameter) and SiNW stiffness and pore size of agglomerates (increasing with diameter). We observe a minimum performance for the mid-sized SiNW_42_: a low specific capacity and a low Coulombic efficiency, which correlates with the formation of a higher quantity of crystalline Li_15_Si_4_ during lithiation. The larger sized SiNW_55_ with a smaller specific area (thus a lower irreversible capacity) and probably a larger size of pores in the agglomerates have a higher specific capacity. On the other hand, the smallest SiNW_9_ offer the highest Coulombic efficiency in the long run, probably due to fast Li diffusion. Additional electrode engineering is, thus, required to take full advantage of the promising performances of SiNWs. An alternative strategy consists of growing SiNWs at the surface of graphite flakes to produce directly a composite anode material [3]. This way, we could reduce micron-sized aggregation and attain full Si capacity cycling. Passivation of the SiNW surface to avoid large irreversible capacity at first cycle is another strategy to explore.

## 5. Conclusions

This paper presents the first extensive comparison of size/shape of nano-silicon (nanoparticles and nanowires) used as anode materials in lithium-ion batteries. The main challenge was to obtain each nanomaterial with a good size and shape control, and in gram-scale quantities for reliable LiB electrochemical tests. We used two independent synthesis processes: laser pyrolysis for SiNPs (now an industrial process) and gold-catalyzed VLS growth in porous media for SiNWs (a scalable patented process). Tuning the synthesis parameters, we obtained SiNPs in a 25–110 nm diameter range. By tuning the gold catalyst size and the porosity of the support, SiNWs with diameters ranging 9–93 nm were realized. An important difference between the two nano-silicon morphologies is that the SiNP material, grown in a gas flow, consists of a fine nanopowder, while the SiNW material, grown in a porous support, consists mostly of 1–10 µm sized agglomerates. Although an advantage in dealing with nano safety, this turns to a disadvantage in the making the electrode, leading to a less homogeneous slurry and to a part of the Si being inaccessible to the electrolyte. Unsurprisingly, the 1D shape appears more difficult than 0D to control and produce at a large scale. 

The matching series of SiNP and SiNW materials was implemented in lithium batteries in the same conditions and cycled the same way. Their electrochemical behavior shows several consistent effects of size and shape. First, the initial specific capacity of the material depends on its shape: it is close to the theoretical 3579 mAh/g for SiNPs, while SiNWs provide an about 25% lower capacity due to the entanglement of SiNWs during the growth. Second, the irreversible specific capacity at the first cycle is linearly correlated to the specific surface area of the materials, thus to the nano-silicon size. Third, the SiNPs prove much more prone to lithiation down to the crystalline Li_15_Si_4_ phase, and probably to electrochemical sintering, than SiNWs. Finally, in the long run, the smallest SiNW electrodes show a better capacity retention, with a Coulombic efficiency above 99.5% after 43 cycles. A size effect is again observed as smaller SiNWs and SiNPs show a lower polarization and, therefore, a better capacity retention. Improvements are currently under investigation, using carbon coating in the case of SiNPs to reduce irreversible capacity and using direct growth on graphite in the case of SiNWs to make more efficient composites.

## Figures and Tables

**Figure 1 nanomaterials-11-00307-f001:**
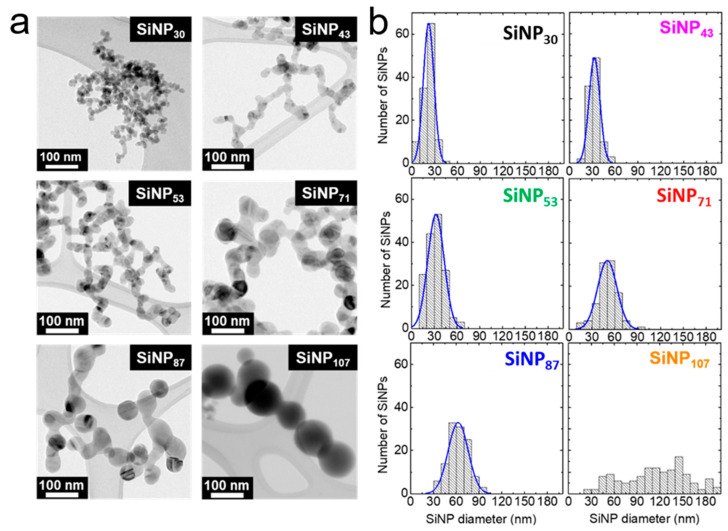
TEM images (**a**) and corresponding size analysis (**b**) of the SiNPs samples described in Table 1. The histograms were obtained from at least 100 particle measurements.

**Figure 2 nanomaterials-11-00307-f002:**
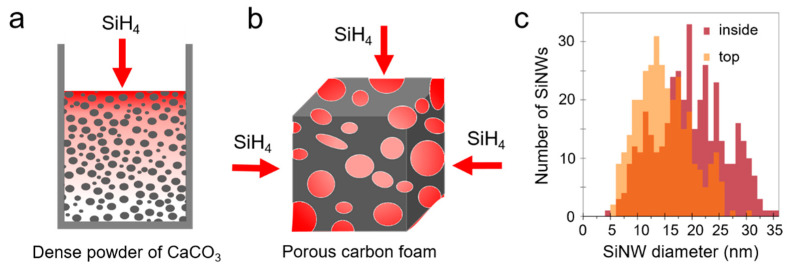
Scheme of the partial pressure reduction in the porous growth support, CaCO_3_ powder (**a**) and carbon foam cube (**b**). The SiH_4_ pressure is represented in red shades. (**c**) Diameters of Si nanowires (SiNWs) grown from 1–2 nm gold nanoparticles (AuNPs) in a 1 cm^3^ carbon foam cube on the surface (“top”) and 2 mm below the surface (“inside”). Inside, SiNWs had the same diameter distribution at all depths down to the center.

**Figure 3 nanomaterials-11-00307-f003:**
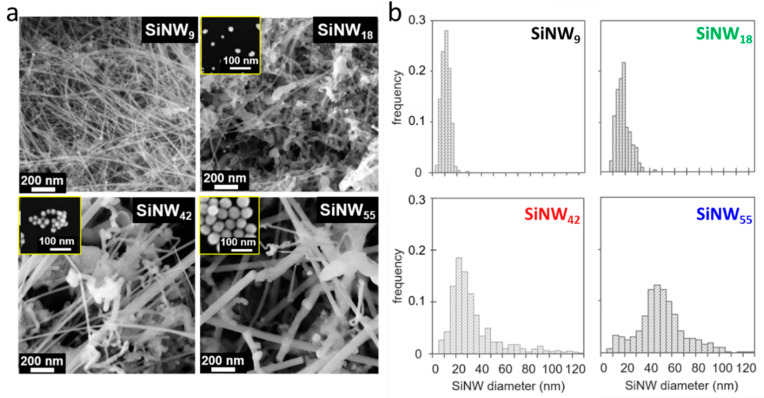
SEM images (**a**) and corresponding diameter histograms (**b**) of SiNWs. Insets: SEM images of the AuNP catalysts used for each SiNW growth. Histograms are calculated on >250 counts.

**Figure 4 nanomaterials-11-00307-f004:**
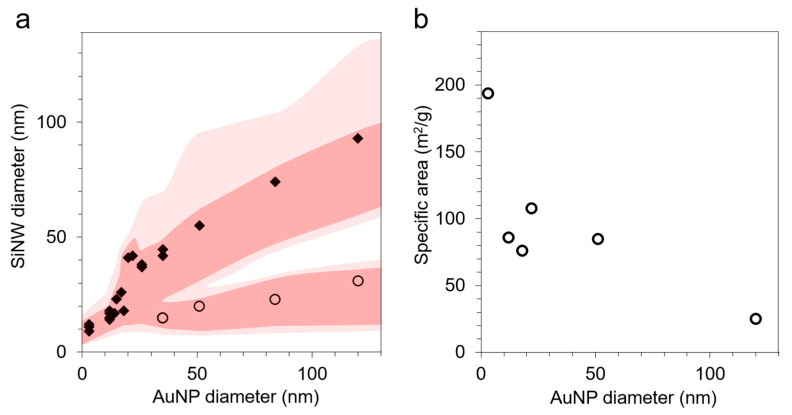
(**a**) Average diameter of the main (diamond) and minority (circles) SiNW populations as a function of the size of the catalyst AuNPs, as measured by SEM (diameter histograms in Figure 3 and Appendix A). The shaded zones show the full width at half maximum and the 90% limit for both SiNW diameter peaks. (**b**) Specific area of the SiNWs as measured by the BET method as a function of the size of the AuNP catalysts.

**Figure 5 nanomaterials-11-00307-f005:**
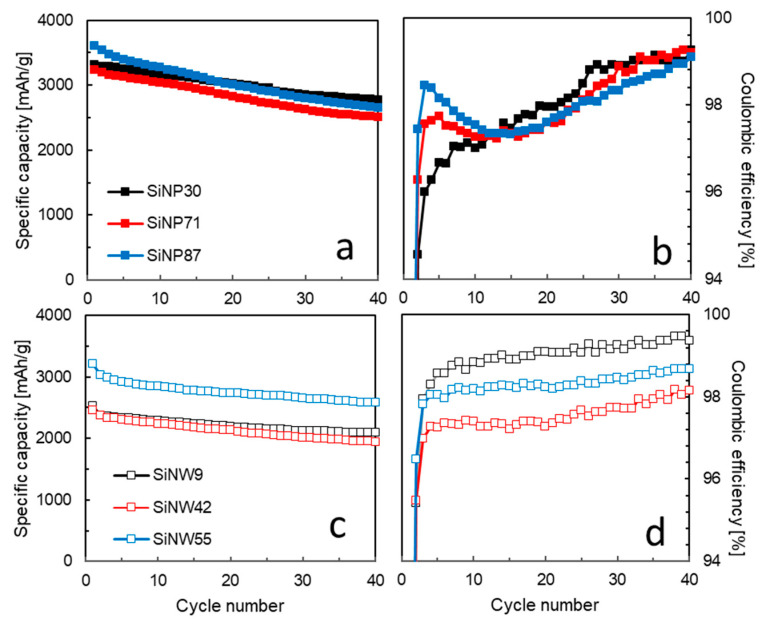
Electrochemical performance of SiNPs and SiNWs measured in half-cell configuration at C/20 rate (activation), then C/5. Specific charge capacity (mAh per gram of silicon) as a function of the cycle number obtained from (**a**) SiNP and (**c**) SiNW electrodes. Corresponding Coulombic efficiency for (**b**) SiNP and (**d**) SiNW electrodes. The results presented here are the most representative of at least 3 repeatable cells (standard deviation 10%). Data for all SiNP sizes are available in Appendix A.

**Figure 6 nanomaterials-11-00307-f006:**
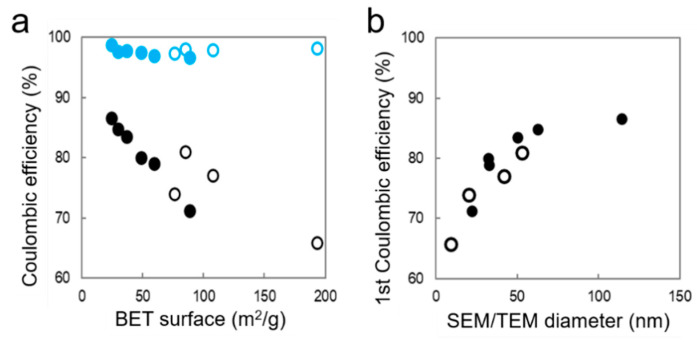
Coulombic efficiency as a function of BET specific area (**a**) and statistical SEM/TEM diameter (**b**) for SiNPs (full circles) and SiNWs (empty circles) on the 1st cycle (black) and the 5th cycle (blue).

**Figure 7 nanomaterials-11-00307-f007:**
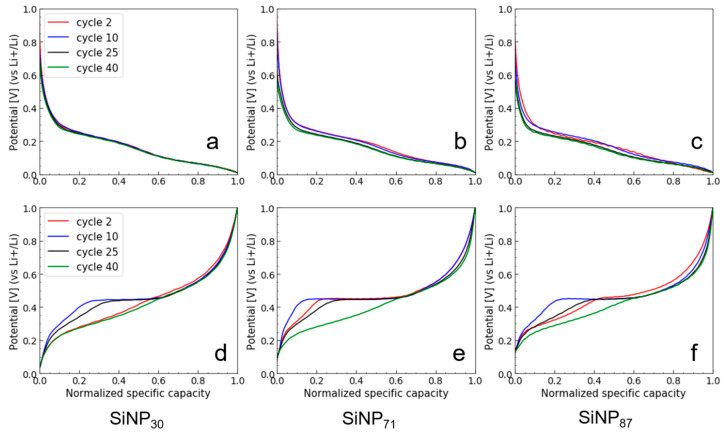
Normalized galvanostatic curves in lithiation (**a**–**c**) and delithiation (**d**–**f**) as a function of the cycle numbers for SiNP_30_ (**a**,**d**), SiNP_71_ (**b**,**e**), and SiNP_87_ (**c**,**f**).

**Figure 8 nanomaterials-11-00307-f008:**
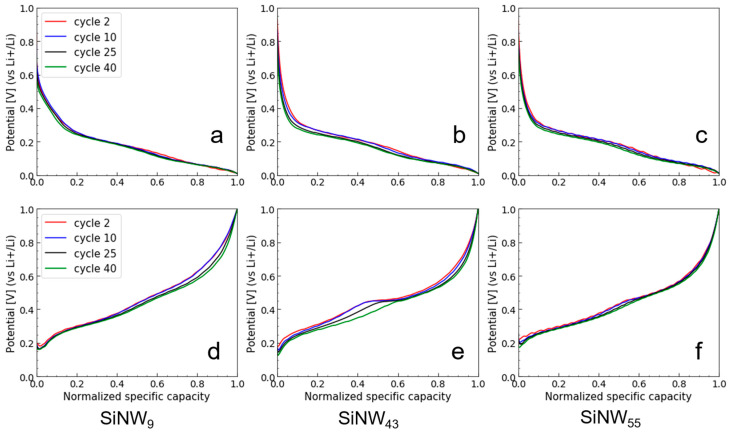
Normalized galvanostatic curves in lithiation (**a**–**c**) and delithiation (**d**–**f**) as a function of the cycle numbers for SiNW_9_ (**a**,**d**), SiNW_42_ (**b**,**e**), and SiNW_55_ (**c**,**f**).

**Figure 9 nanomaterials-11-00307-f009:**
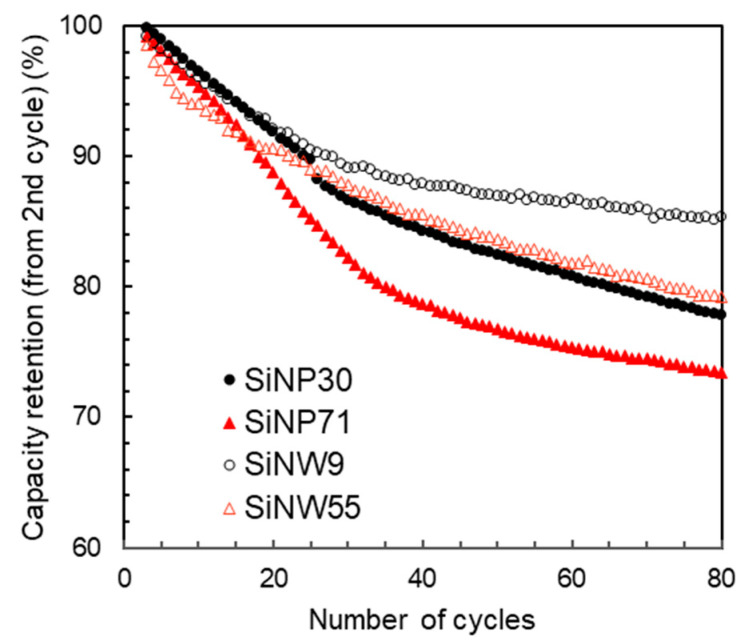
Capacity retention for small (black circles) and large (red triangles) SiNPs (full symbols) and SiNWs (empty symbols).

**Table 1 nanomaterials-11-00307-t001:** Main experimental parameters and characterizations of the Si nanoparticles (SiNP) samples.

Sample	SiH_4_ (sccm)	BET Specific Area (m^2^/g)	Diameter (nm)		Crystallite Size (nm) from XRD
			From BET	From TEM	
SiNP_30_	50	88	29	22	6
SiNP_43_	100	60	43	33	25
SiNP_53_	100	49	53	32	18
SiNP_71_	150	36	71	50	35
SiNP_87_	200	30	87	62	44
SiNP_107_	234	24	107	114	63

**Table 2 nanomaterials-11-00307-t002:** Characteristics of the SiNW samples.

Sample	AuNP Size (nm)	BET Specific Area (m^2^/g)	Diameter from SEM (nm)	
			Small Population	Large Population
SiNW_9_	1–2	194	-	9
SiNW_18_	12	86	-	18
SiNW_20_	18	76	-	20
SiNW_42_	22	108	20	42
SiNW_55_	51	85	23	55
SiNW_93_	120	25	31	93

## Data Availability

The data presented in this study are available on request from the corresponding author.

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
