# Peer review of "Effect of Size and Shape on Electrochemical Performance of Nano-Silicon-Based Lithium Battery"

_nanomaterials, 2021, doi:10.3390/nano11020307_

Round 1

Reviewer 1 Report

This paper reports fabrication and electrochemical properties of nano-Si  for lithium battery anodes. Interesting results were found. I  recommend accept of the manuscript after following revisions:

(1) Find a colleague to revise many grammatical errors in the whole text.

(2) The title could be revised by adding focused property area such as "electrochemical property".

Author Response

Reviewer 1

Open Review

This paper reports fabrication and electrochemical properties of nano-Si  for lithium battery anodes. Interesting results were found. I  recommend accept of the manuscript after following revisions:

(1) Find a colleague to revise many grammatical errors in the whole text.

A correction by a native English speaker has been made.

(2) The title could be revised by adding focused property area such as "electrochemical property".

The title has been changed for more clarity to “Effect of size and shape on electrochemical performance of nano-silicon based lithium battery”.

Reviewer 2 Report

Surely nano silicon is a good anode material for lithium ion batteries, and there are some practical applications. In this ms, it studied the SiNPs and SiNWs and found there is some relationship between initial Coulomb efficiency and size. It is interesting to publish after revision:

  1. There are some pioneering works such as T. Zhang et al. published in J. Mater. Chem. and Electrochem. Commun. The authors should make not some of them instead of some current references. 
  2. The authors should be careful prior to the submission. For example, some "Error! 142 Reference source not found." appeared not just once in the text.
  3. The Coulomb efficiency is not high, and their use for practical application is impossible. Please supplement some methods in the discussion for further improvements. 
  4. The reasons why SiNWs are superior to SiNPs in cycling should be expounded. 
  5. The amount of conductive agent is too much, which also contributes some capacity. Please evaluate it. 

Author Response

Reviewer 2

Open Review

Surely nano silicon is a good anode material for lithium ion batteries, and there are some practical applications. In this ms, it studied the SiNPs and SiNWs and found there is some relationship between initial Coulomb efficiency and size. It is interesting to publish after revision:

  1. There are some pioneering works such as T. Zhang et al. published in J. Mater. Chem. and Electrochem. Commun. The authors should make not some of them instead of some current references. 

The reference was added as ref. 5.

  1. The authors should be careful prior to the submission. For example, some "Error! 142 Reference source not found." appeared not just once in the text.

Sorry, there was a problem in turning the docx manuscript into pdf at the submission. This is now corrected.

  1. The Coulomb efficiency is not high, and their use for practical application is impossible. Please supplement some methods in the discussion for further improvements. 

Indeed, the study is focused on comparing silicon nanomaterials on their shape and size, and thus the materials were compared in the exact same conditions, that could not be optimized for all of them. Optimizing the performance is the aim of parallel studies, some of which have been published recently. You are right; the methods for improvements are worth citing here. We added statements on lines 389-391 for SiNPs and 407-412 for SiNWs:

“Alternatively, sintering is efficiently prevented by a carbon coating, conveniently obtained in the same reactor just after pyrolysis, as we recently reported.[60]”,

“Additional electrode engineering is thus required to take full advantage of the promising performances of SiNWs. An alternative strategy consists in growing SiNWs at the surface of graphite flakes to produce directly a composite anode material.[3] This way, we could reduce micron-sized aggregation and attain full Si capacity cycling. Passivation of the SiNW surface to avoid large irreversible capacity at first cycle is another strategy to explore.”.

  1. The reasons why SiNWs are superior to SiNPs in cycling should be expounded. 

SiNPs and SiNWs have their own advantages. To make a long story short, SiNPs are superior in short cycling and SiNWs in long cycling. We changed the text in several places to make this clearer to the reader. We added a paragraph 392-298 specifically dedicated to the comparison of the electrochemical performance as a function of the shape:

“When comparing SiNWs to SiNPs, it appears that the initial specific capacity of SiNWs is lower due to their micron-sized agglomerate form, while the CE of SiNWs is more stable and can be very high for the smallest size. It seems that the 1D shape of SiNWs brings a significant advantage in maintaining a 3D porous structure in the ma-terial, because stiff wires do not pack as tightly as 0D particles. This 3D structure has a strong impact both on lithium diffusion and on avoiding Si electrochemical sintering: much less crystalline phase Li15Si4 forms in cycling.”.

The conclusion summarizes with several comparison points on lines 430-439:

“The matching series of SiNP and SiNW materials was implemented in lithium batteries in the same conditions, and cycled the same way. Their electrochemical be-havior shows several consistent effects of size and shape. First, the initial specific ca-pacity of the material depends on its shape: it is close to the theoretical 3579 mAh/g for SiNPs, while SiNWs provide an about 25 % lower capacity due to the entanglement of SiNWs during the growth. Second, the irreversible specific capacity at the first cycle is linearly correlated to the specific surface area of the materials, thus to the nano-silicon size. Third, the SiNPs prove much more prone to lithiation down to the crystalline Li15Si4 phase, and probably to electrochemical sintering, than SiNWs. Finally, in the long run, the smallest SiNW electrodes show a better capacity retention, with a Cou-lombic efficiency above 99.5 % after 43 cycles. A size effect is again observed as smaller SiNWs and SiNPs show a lower polarization and therefore a better capacity retention.”.

  1. The amount of conductive agent is too much, which also contributes some capacity. Please evaluate it. 

The contribution of the conductive agent to the capacity was measured independently. The result has been added on line 275-276:

“The contribution of the carbon black to the specific capacity, measured independently, is a constant 100mAh/g.”.

Reviewer 3 Report

141: Reference to Table 1 is missing
156: Fig 1 is added two times to the manuscript
199: Typo: 2x "locally the"
203 and 210: Fig 2 is added three times to the manuscript
211: Ref sould be to Fig 2c not to Fig 2b
218: In Fig 3, it is not indicated which part is "a" and which part is "b"
268: Fig 5: Chose same colour code for SiNP and SiNW graphs. For example blue = thinnest/ red = midel size / black = largest size
274: Reference to Fig 3 is missing
279: "The lower specific capacity of the SiNWs can be explained by the more difficult dispersion of the material during the slurry preparation". Why is the capacity of the SiNW55 higher? Are there more/larger agglomerates?
311: "impact of size reduction is clearly visible on the SiNWs". I do not see a clear/logic trend ifo of SiNW size in Fig 3, nor for capacity ifo cycles neither for the Coulombic efficiency ifo cycles. For capacity ifo cycles, the thickest, SiNM55, performs best. For Coulombic efficiency ifo of cycles, the thinnest, SiNW9 performs best. If thinner is better for Coulombic efficiency vs cycles, why is SiNW55 better than SiNW 42? Please comment/clarify in the manuscript.
332: "This process is less visible for the SiNWs (Figure 8) indicating again an influence of the Si shape on the Li diffusion." Indeed for SiNWs there is no increase in polarization when increase diameter and there is no increase in polarization when cycling. Nevertheless the polarization seems to be at the same level as the SiNPs (Fig S8). Could it be that another effect (slurry compoistion, electrode porosity,...) is affecting overpotential and hiding any impact from diameter and from cycling? Please comment.
341: "According to the literature, this phase 341 also reduces the Coulombic efficiency,[58] which is in agreement with the observed CE evolution of the SiNPs in our study". I assume the observed CE evolution refers to Fig 5b? In Fig 5b, the Ce drops between cycle 5 - cycle 15. In line 315, it is claimed that this decline is attributed to electrochemical sintering and pulverisation. Are these two effects (electrochemical sintering or Li15Si4 formation) the same? If not, which of the two effects is dominant. Please clarify and supply supporting information.

Author Response

Reviewer 3

Open Review

141: Reference to Table 1 is missing
156: Fig 1 is added two times to the manuscript
199: Typo: 2x "locally the"
203 and 210: Fig 2 is added three times to the manuscript

We apologize for the bad shape of the submitted pdf file. There was a software error in processing the pdf file from the docx original document. It is now corrected.

211: Ref sould be to Fig 2c not to Fig 2b
218: In Fig 3, it is not indicated which part is "a" and which part is "b"
268: Fig 5: Chose same colour code for SiNP and SiNW graphs. For example blue = thinnest/ red = midel size / black = largest size
274: Reference to Fig 3 is missing

Thank you for this thorough reading. Corrections have been made accordingly.

279: "The lower specific capacity of the SiNWs can be explained by the more difficult dispersion of the material during the slurry preparation". Why is the capacity of the SiNW55 higher? Are there more/larger agglomerates?

The reason why the capacity of SiNW55 is higher is not clearly identified and it will require deeper investigations. As mentioned, the capacity measurements on SiNWs are less repeatable than on SiNPs due to issues in agglomerate dispersion. The higher capacity of SiNW55 may be related to better wettability than for SiNW42, but we do not have any evidence by microscopy. Whatever the capacity, the capacity retentions are similar and the best performing SiNWs can be chosen from Coulombic efficiency. We thus introduced a more detailed discussion of the relative capacities of SiNW samples on paragraph 278-282:

” The specific capacity is higher and more repeatable for SiNPs (3000-3500 mAh/g) than for SiNWs (2500-3000 mAh/g). All SiNP anodes give a similar specific capacity. For SiNWs, we observe a minimum of specific capacity for the medium size, SiNW42, and a best trade-off between specific capacity and first irreversible capacity for the largest, SiNW55 (Figures 5c and S6). However, among all samples, only the smallest SiNW anode, SiNW9, attains the Coulombic efficiency of 99.5% required for long term cycling (figure 5d).”.

311: "impact of size reduction is clearly visible on the SiNWs". I do not see a clear/logic trend ifo of SiNW size in Fig 3, nor for capacity ifo cycles neither for the Coulombic efficiency ifo cycles. For capacity ifo cycles, the thickest, SiNM55, performs best. For Coulombic efficiency ifo of cycles, the thinnest, SiNW9 performs best. If thinner is better for Coulombic efficiency vs cycles, why is SiNW55 better than SiNW 42? Please comment/clarify in the manuscript.

You are right, this sentence has been removed. There is an influence of size on the electrochemical performances of SiNWs, but it does not show a simple trend. We think that the lower Coulombic efficiency in SiNW42 is linked to its making more crystalline Li15Si4 in lithiation than the other samples. It is clear that we do not see a linear trend such as “thinner is better”, but a complex trade-off that will require deeper investigations that do not fit into this already long paper. A more detailed discussion of this point has been added on paragraph 356-366:

“Surprisingly this crystalline Li15Si4 delithiation process is much less visible in the SiNW electrodes, with very small potential plateaus at 450 mV (Figure 8d-f). This shows that the crystalline Li15Si4 phase is present in much smaller quantity, or that the lithiated phase remains mostly amorphous.[53] Only the mid-sized SiNW42 show a significant plateau, a fact that can be correlated with its lowest Coulombic efficiency as compared to the other SiNW samples (Figure 5d). A reason why SiNWs do not undergo electrochemical sintering as easily as SiNPs do, might be related to their elastic “spring” behavior. Even after grinding and calendaring, SiNWs in the anode remain stiff and maintain distances. Agglomerates of SiNWs thus contain a nanoscale porosity that was clearly imaged in our recent FIB-SEM study.[3] Such distance allows swelling in lithiation with a low risk of sintering with neighboring SiNWs.”.

332: "This process is less visible for the SiNWs (Figure 8) indicating again an influence of the Si shape on the Li diffusion." Indeed for SiNWs there is no increase in polarization when increase diameter and there is no increase in polarization when cycling. Nevertheless the polarization seems to be at the same level as the SiNPs (Fig S8). Could it be that another effect (slurry compoistion, electrode porosity,...) is affecting overpotential and hiding any impact from diameter and from cycling? Please comment.

The slurry composition is the same, the electrode porosity is high in all cases (50-90%), they can not explain the differences in polarization. This point is still under study. The sentence was unclear, and the paragraph 330-336 was modified accordingly:

” During the first lithiation (Figure S8), the very long potential plateau at ca. 100mV is shifted toward lower potential for larger SiNPs, indicating polarization. This is in agreement with a slower Li diffusion along longer distances within large SiNPs.[52] In the following cycles (Figure 7), we can see that a polarization in lithiation is slightly building up as a function of the cycling number for the large SiNPs. For the SiNWs, the potential plateau in the first cycle is also low, indicating anode polarization, but it does not depend on size (Figure S8), and in the subsequent cycles, polarization does not increase (Figure 8). This discrepancy indicates an influence of the Si shape on the Li diffusion.”.

341: "According to the literature, this phase 341 also reduces the Coulombic efficiency,[58] which is in agreement with the observed CE evolution of the SiNPs in our study". I assume the observed CE evolution refers to Fig 5b? In Fig 5b, the Ce drops between cycle 5 - cycle 15. In line 315, it is claimed that this decline is attributed to electrochemical sintering and pulverisation. Are these two effects (electrochemical sintering or Li15Si4 formation) the same? If not, which of the two effects is dominant. Please clarify and supply supporting information.

Yes, from the literature, we inferred that the electrochemical sintering, the subsequent pulverization and the formation of crystalline Li15Si4 are indeed different consequences of deep lithiation in Silicon anodes. It is difficult to state which effect is dominant. The formation of crystalline Li15Si4 shows that we have a deep lithiation, and it adds an additional volume change increasing pulverization. It probably facilitates the sintering, and the sintering leads to particles larger than the critical pulverization size. We added a sentence 386-388 in the discussion explaining this hypothesis:

“Polarization, crystalline Li15Si4 formation and lower CE might be attributed to a large scale phenomenon of electrochemical sintering.[49]”.

The discussion was fully reorganized to be more explicit around this question.
